# Sensing through Non-Sensing Ocular Ion Channels

**DOI:** 10.3390/ijms21186925

**Published:** 2020-09-21

**Authors:** Meha Kabra, Bikash Ranjan Pattnaik

**Affiliations:** 1McPherson Eye Research Institute, University of Wisconsin-Madison, Madison, WI 53705, USA; mkabra2@wisc.edu; 2Department of Pediatrics, University of Wisconsin-Madison, Madison, WI 53706, USA; 3Department of Ophthalmology and Visual Sciences, University of Wisconsin-Madison, Madison, WI 53705, USA

**Keywords:** ocular ion channels, inherited channelopathy, disease models, nonsense mutation therapies, CRISPR-DNA/RNA editing, readthrough, anticodon engineered tRNA

## Abstract

Ion channels are membrane-spanning integral proteins expressed in multiple organs, including the eye. In the eye, ion channels are involved in various physiological processes, like signal transmission and visual processing. A wide range of mutations have been reported in the corresponding genes and their interacting subunit coding genes, which contribute significantly to an array of blindness, termed ocular channelopathies. These mutations result in either a loss- or gain-of channel functions affecting the structure, assembly, trafficking, and localization of channel proteins. A dominant-negative effect is caused in a few channels formed by the assembly of several subunits that exist as homo- or heteromeric proteins. Here, we review the role of different mutations in switching a “sensing” ion channel to “non-sensing,” leading to ocular channelopathies like Leber’s congenital amaurosis 16 (LCA16), cone dystrophy, congenital stationary night blindness (CSNB), achromatopsia, bestrophinopathies, retinitis pigmentosa, etc. We also discuss the various in vitro and in vivo disease models available to investigate the impact of mutations on channel properties, to dissect the disease mechanism, and understand the pathophysiology. Innovating the potential pharmacological and therapeutic approaches and their efficient delivery to the eye for reversing a “non-sensing” channel to “sensing” would be life-changing.

## 1. Introduction

Ion channels are membrane-spanning transport proteins that facilitate the passive bidirectional (into and out of cell and cell organelles) flow of selective ions like sodium (Na^+^), potassium (K^+^), calcium (Ca^2+^), chloride (Cl^−^), or unspecific cations (Figure 1). These ion channel proteins have an enormous heterogeneity in their electrophysiological properties. In the eye, numerous ion channels are found at different anatomical locations like cornea, lens, ciliary body, and retina, as outlined in Figure 2. A flux of ions generates a membrane potential that maintains the ionic homeostasis required for the proper functioning of cells like signal transmission and visual processing in ocular cells [1,2,3,4]. Ion channels are either gated or non-gated, which permit selective ions to pass through the membrane based on the electrochemical gradient. Gated ion channels are regulated by electrical (membrane potential, voltage), chemical (ligands, cyclic nucleotides, phosphorylation), or other signals (light, temperature, pH, or mechanical stimuli). In contrast, non-gated channels allow the free flow of ions across the membrane.

The steady-state regulation of ionic balance by channel proteins is a critical phenomenon in the eye for most of the cellular functions. The most common is the cyclic nucleotide-gated (CNG) cationic channel, which mediates phototransduction in rod and cone photoreceptors by regulating ligand-dependent homeostasis (Na^+^ and Ca^2+^) [5,6,7,8,9]. Ca^2+^ homeostasis is also required to regulate the physiology of the lens, and an overload of Ca^2+^ is detrimental to the lens, leading to cortical cataract [10]. Different combinations of channels (voltage-operated Ca^2+^ channels; VOCCs, receptor-operated Ca^2+^ channels; ROCCs, second messenger-operated Ca^2+^ channels; SOCE, transient receptor potential channels; TRPs) tightly regulate the Ca^2+^ influx in lens to maintain its transparency [10,11]. Voltage and ligand operated Ca^2+^, K^+^, Cl^−^ channels in the retinal pigmented epithelial (RPE) layer contribute to the secretory activity, volume regulation, and transepithelial ion transport to maintain the retinal health. In contrast, alterations in these channel functions lead to retinal degeneration [12]. K^+^ channel activity modulates essential action potential in excitable cells and transport and secretory activity in non-excitable cells. K^+^ channels (inwardly rectifying K^+^ (K_ir_)-channels, voltage-gated K^+^ channels (K_v_), Ca^2+^-activated K^+^ channels, two-pore or leak K^+^-channels) are imperative in ocular tissues like the cornea to regulate epithelial cell proliferation and apoptosis [13,14,15], the lens to maintain the volume and transparency [16], the retinal ganglion cells to modulate the resting membrane potential and cell excitability [17], and the RPE cell physiology, for its interaction with the photoreceptor and ionic composition of the subretinal space [18,19]. The passive flow of Cl^−^ in ocular tissues like cornea, conjunctiva, ciliary epithelium, and RPE is mediated by Cl^−^ channels (high- (maxi) conductance, the cystic fibrosis transmembrane conductance regulator (CFTR), volume-regulated, voltage-gated, Ca^2+^-activated) to regulate tear film volume, aqueous humor volume, corneal transparency, and ionic composition [20,21,22,23,24,25,26,27,28,29,30].

The human genome has over 400 ion channel genes, and several mutations have been reported across several ion channel genes leading to various pathological conditions, known as channelopathies [31]. These mutations may alter the structure, assembly, localization, trafficking, or functions of channel protein and, therefore, turn a sensing ion channel into a non-sensing one (Figure 1C). The effect of these mutations can be studied by electrophysiological techniques. Gene mutations can result in a loss or gain of functions. Structurally, these channel proteins are formed by the assembly of several similar (homomeric) or different (heteromeric) subunits. Therefore, loss-of-function mutations sometimes result in a dominant-negative effect in ion channels. Most of the ocular channelopathies are rare conditions and have been a rapidly expanding, yet unexplored area of ophthalmology. There is a substantial amount of genetic and allelic heterogeneity in channelopathies. It remains unclear how different mutations in the same gene can result in a wide range of phenotypic variability. For example, *BEST1* gene mutations result in a spectrum of ocular phenotype such as microcornea, cataract, retinitis pigmentosa, and macular and rod-cone dystrophy [32]. Interestingly, mutations in the same gene exhibit a different inheritance pattern. For example, mutations in the *KCNJ13* gene lead to LCA16, an autosomal recessive disease, and snowflake vitreoretinal degeneration (SVD), an autosomal dominant disease [33,34,35]. 

Many ion channel genes are expressed in the eye, and few are known to harbor the disease-causing mutations. These genes involved in ocular ion channelopathies were identified by chromosomal mapping and targeted exome sequencing (*KCNJ13*; [35]), microsatellite analysis (*CACNA1F*; [36]), polymorphism analysis, and recombination mapping (*BEST1*; [37]), based on the phenotype in a mutant mouse model (*CACNA2D4*; [38]) and, homozygosity mapping and linkage analysis (*KCNV2* [39], *CNGA3* [40], *CNGB3* [41], *CNGB1* [41], *CNGA1* [42], *TRPM1* [43]). The present article reviews the fundamentals of ocular ion channelopathies caused by these genes, with an exclusive insight into the role of genetic mutations in disease pathogenesis. We also discuss the future percepts of potential pharmacological and therapeutic strategies.

## 2. Genetic Mutations Switching the “Sensing” Ion Channels to “Non-Sensing” Ones

Various mutations that have been observed so far across ocular ion channel genes are illustrated in Figure 3. Each channelopathy has its genetic signature, which makes the screening of disease-causing genes crucial. Most of the mutations observed across these genes are missense, followed by frameshifts, nonsense, and splice-site mutations. Missense mutations in ion channel genes result in a single amino acid change, which is sufficient to alter the assembly/trafficking/function of proteins. The best example is a novel missense variant (p.C319R) in *CNGA3* identified by whole-exome sequencing in a family with cone-rod dystrophy (CRD). The mutation is located at the fifth transmembrane domain of the protein, which is essential for the structural and functional integrity of the cone CNG channel. The mutant protein has significantly diminished channel activity due to reduced stability, misfolding, and altered trafficking to the membrane [44]. Some of the missense mutations also result in a dominant-negative or gain-of-function effect on channel functions, while some remain unknown for their role in disease mechanism. Biallelic mutations in the *CACNA2D4* gene have been reported in patients with autosomal recessive cone-rod dystrophy, while carriers with heterozygous mutation did not develop any disease phenotype [45,46]. A 30 kb heterozygous deletion (including exon 19 to 26) in the *CACNA2D4* gene with a homozygous deletion in *C21orf2* was observed in a patient with retinal degeneration (RD) who clinically had a similar phenotype as an RD-patient with only homozygous *C21orf2* mutation, suggesting the noninvolvement of *CACNA2D4* in RD [47]. 

### 2.1. Contribution of Nonsense Mutations in Ocular Channelopathies

Nonsense mutations are the most deleterious as they result in either no or truncated protein production because of premature termination codon (PTC). PTC occurs when a canonical genetic codon abruptly gets converted into one of the three stop codons (TAG, TGA, or TAA). PTCs account for 10–15% of all genetic diseases [48,49]. Several ocular channelopathies have been reported (Table 1) to be associated with nonsense mutations including LCA16 [33], cone dystrophy [46,50,51], CSNB [52], bestrophinopathies [53] achromatopsia [54], and retinitis pigmentosa [55]. Table 1 represents the number of nonsense mutations reported across different disease-causing genes (HGMD database; http://www.hgmd.cf.ac.uk/ac/index.php). The effect of PTCs is pronounced when nonsense-mediated decay (NMD) alters the mRNA abundance. NMD is a post-transcriptional regulation by mRNA surveillance process, which can degrade the aberrant mRNA carrying PTC as it may be harmful to the physiological functioning and prevent the formation of non-functional protein [56]. Substitutions and frameshift mutations (p.Glu80X, p.Gln223X, and p.Asp154Ala fsx58) in *KCNV2* leading to PTC would lack all the transmembrane domains and P-loop of the channel. If it escapes NMD, the polypeptide would be completely mislocalized and non-functional (Vincent et al., 2013). NMD processing of only a few ocular ion channel PTC mutations has been studied. One such example is *BEST1* mutations (c.521_522del and c.1100+1G>A) in patients with autosomal recessive bestrophinopathy that result in PTC and undergo NMD [57]. A mechanism of PTC with and without NMD leading to inherited blindness is illustrated in Figure 4.

### 2.2. TRPM1 (Transient Receptor Protein Melastatin 1; MLSN1)

The gene contains 28 exons that encode for the TRPM1 protein of 1625 amino acids (NM_001252024.2). The channel protein has a vital role in light-induced depolarization of ON-bipolar cells. The channel comprises six transmembrane domains with a pore region between the last two domains, a coiled-coil region, an intracellular N-terminal domain with conserved residues, and an intracellular C-terminal domain containing a TRP motif [58]. TRPM1 is a nonselective cation channel with preference to Ca^2+^ flux, but its larger isoforms (>1625 up to 1643 amino acids) prefer Na^+^ over Ca^2+^ [59]. In the eye, the channel is expressed in the dendrites of ON bipolar cells communicating with rod photoreceptors and mediates the synaptic transmission. The channel is synchronously regulated by mGluR6 signaling [60,61,62]. A visual signal is segregated into ON and OFF pathways by retinal bipolar cells for visual processing [63]. In the dark, photoreceptor cells are depolarized as Na^+^ and Ca^2+^ channels remain open due to high levels of intracellular cGMP. The flux of Na^+^ ions is counterbalanced by Na^+^/K^+^ pumps at the expense of ATP. Subsequently, the neurotransmitter glutamate is released from the photoreceptor synaptic vesicles, which binds to its receptor (mGluR6 in bipolar cells) and activates the heterotrimeric G_0_ complex. This negatively modulates the TRPM1 channel and the closing of the channel results in the hyperpolarization of ON-bipolar cells [60,64]. In light, photoreceptor cells are hyperpolarized due to decreased cGMP levels, represented as electroretinogram (ERG) a-wave. The closing of Na^+^ and Ca^2+^ channels reduces glutamate concentration in the synapse and depolarizes the ON-bipolar cells, ERG b-wave [65,66,67,68,69]. Mutations in *TRPM1* gene results in autosomal-recessive complete congenital stationary night blindness (cCSNB or CSNB1) [52,62,70,71,72,73]. These patients, along with exhibiting nystagmus, myopia, and amblyopia, exhibit a reduced or completely absent ERG-b wave, similar to the Schubert–Bornschein patient ERG response [74], due to defective ON-bipolar cells. Most of the patients with cCSNB have functional photoreceptors, as represented by normal ERG a-wave. Mutations, based on types and locations, can alter the physiological function of the TRPM1 channel. Missense mutations at the N-terminal (i.e., R624C) and C-terminal (F1075S) of TRPM1 alter the localization of channel protein to the dendritic tips of bipolar cells. TRPM1 is also mislocalized due to mutations in other CSNB genes like nyctalopin (*NYX*), which encodes a protein required for its proper translocation. Nonsense mutations (i.e., S882X) in TRPM1 result in no protein product and hence the absence of channel in ON bipolar cells leading to CSNB (Figure 5). Similarly, splicing variants (i.e., IVS2–3C>G, IVS8+3_6delAAGT) also result in an abnormal protein product with a loss of function effect [73]. 

### 2.3. CACNA1F (Voltage-Gated Calcium Channel Alpha-1F Subunit)

The gene contains 48 exons encoding the L-type Ca^2+^ channel protein subunit, Ca_v_1.4α1 of 1966 amino acids (NM_001256789.3). The primary pore-forming and voltage sensing subunit, α1, co-assembles with other auxiliary β and α_2_δ subunits to form the voltage-gated Ca_v_1.4 channel [75]. β and α_2_δ subunits are crucial for the trafficking of the α1 subunit to the membrane and shielding it from proteasomal degradation [76,77,78,79]. The Ca_v_1.4 channel is present in synaptic terminals of rods and cone photoreceptors and is vital for Ca^2+^ mediated neurotransmission. The channel is also expressed in the inner nuclear layer, outer nuclear layer, and ganglion cell layer of the retina [80]. Ca^2+^ influx via the Ca_v_1.4 channel in the photoreceptor prompts the glutamate release from the synaptic ribbon of rods and cones to transmit the signals to bipolar cells [81]. Mutations in *CACNA1F* cause X-linked incomplete congenital stationary blindness (CSNB2 or iCSNB), resulting in night vision impairment, myopia, strabismus, nystagmus, and reduced (or no) ERG b-wave [36,71,82,83,84,85,86]. These patients manifest functional defects in rod and cone photoreceptors, unlike CSNB1 patients (which occur due to mutations in *TRPM1*, *NYX*, *GRM6,* etc.) with only rod related problems [87]. Pathogenic mutations in *CACNA1F* perturb the channel function and result in a diminished influx of Ca^2+^ ions leading to reduced release of glutamate [88]. In the retina synapse, as stated earlier, glutamate levels are controlled during light-dark transitions [89]. A low level of glutamate due to mutated Ca_v_1.4α1 upsets these transitions [89,90]. The mutant *Cacna1f*-loss-of-function mice showed the disruption of cellular organization in second-order neurons, ribbon synapses at the outer plexiform layer, and Ca^2+^ signaling, leading to perturbed retinal neurotransmission [91]. A missense mutation, I745T, resulted in increased channel activity with slower inactivation kinetics [92]. Allelic heterogeneity is stated in *CACNA1F*, in which different mutations in a single gene lead to a distinct phenotype. The *CACNA1F* mutations also cause cone-rod dystrophy (CORD), which is a monogenic disease characterized by cones that are more severely affected than the rod photoreceptor function. A large in-frame deletion in *CACNA1F*-exons 18 to 26 in a German patient was reported to be associated with CORDX3 [93,94,95]. An intronic-exonic deletion of 425bp encompassing exon 30 and adjacent introns of *CACNA1F* resulted in the deletion of the TM domain and extracellular loop of Ca_v_1.4α1 in a patient with Aland Island eye disease (AIED), a nearly identical disorder to CSNB2 [96]. 

### 2.4. CACNA2D4 (Voltage-Gated Calcium Channel Alpha-2/Delta Subunit 4)

The gene contains 38 exons, which encode an auxiliary subunit α_2_δ4 of an L-type voltage-gated Ca^2+^ channel (Ca_v_1.4) comprised of 1137 amino acids (NM_172364.5). The Ca_v_1.4α2δ4 protein is mainly expressed in the photoreceptor and outer plexiform layer containing photoreceptor synapses [75,97]. The regulated influx of Ca^2+^ at synaptic terminals of rods and cones through this channel is essential to trigger the glutamate release for synaptic transmission to multiple postsynaptic targets and vision [89,98]. The channel is also involved in maintaining the structure, molecular composition of synapses, and exocytosis at mature synaptic ribbons of these cells [99]. The disruption of channel functions due to mutations leads to cone-mediated heterogeneous vision impairment, known as autosomal recessive cone dystrophy [46,100,101]. The disease was first noticed in two siblings who had vision loss problems with reduced scotopic ERG and subnormal cone response. Both the siblings had a homozygous nonsense mutation (c.2406C>A, Tyr802X) in *CACNA2D4,* truncating 33% of the respective ORF [46]. The mouse model of X-linked congenital stationary night blindness exhibited a lack of synaptic ribbons with increased Ca^2+^ influx [102,103,104]. Similarly, in mice with a nonsense mutation in *Cacna2d4,* structural and functional disruptions of photoreceptors were observed [46,105,106]. Nonsense mutations in the Ca_v_1.4 produce a short splice variant that is incapable of increasing the presynaptic surface density of Ca_v_ channels, unlike a full-length option leading to either loss of functions or abnormal photoreceptors [75,107]. Channel insufficiency in knock-out mice caused the disruption of photoreceptor’s synaptic structures and altered presynaptic Ca^2+^ signaling and functions [91,100,108,109]. The mutated α2δ4 altered the structure of synaptic terminals in photoreceptors, which subsequently affected the synaptic protein spatial arrangement leading to loss of signal transmission as indicated by diminished or lost ERG b-wave in *Cacna2d4* mutant mice with a nonsense mutation. The Ca_v_1.4 channel, via its α1 subunit, interacts with other channels like TMEM16 (Ca^2+^ activated Cl^−^ channel) located in synaptic terminals of photoreceptors and regulates their activity, which in turn control the Ca_v_1.4 channel by a feedback mechanism. The TMEM channel is susceptible to altered Ca^2+^ influx from a mutated Ca_v_1.4 and no longer present in the disorganized synaptic structures in the outer plexiform layer (OPL), rather delocalized to the cell body in the outer nuclear layer (ONL). Ca^2+^-dependent chloride current (I_Cl(Ca)_) was nearly abolished in these mice, indicating the defective TMEM16 trafficking and channel functions [106].

### 2.5. KCNV2 (Voltage-Gated K^+^ Channel)

The *KCNV2* gene with 2 exons encodes for the K_v_8.2 channel subunit of 545 amino acids (NM_133497.4), expressed in the inner segment of the photoreceptor layer of the retina [110]. Structurally, the channel is composed of six transmembrane domains, of which the fourth is positively charged due to Arg or Lys at every fourth position that acts as a principle voltage sensor. It also contains an N-terminal A and B box (T1 domain) and a P-loop (pore-forming loop) present between the last two domains [111,112]. KCNV2 exists as a functional hetero-tetrameric complex with subunits of other K^+^ channels like K_v_2.1 [110,111,113]. The assembly of K_v_8.2 with K_v_2.1 (*KCNB1*) results in a permanent outward K^+^ current in the photoreceptors with lower membrane potential required for activation and reduced deactivation kinetics [110]. Recessive mutations in *KCNV2* have been associated with cone dystrophy with supernormal rod responses (CDSRRs) [39,50,114,115]. Patients with CDSRR present with photophobia, nyctalopia, reduced color perception, macular changes in later stages of the disease, and reduced and delayed cone response in addition to supernormal ERG b-wave [116,117,118,119,120,121]. The most commonly observed mutation in *KCNV2* was G461R, in the third amino acid of highly conserved ion-selective filter forming tripeptide (Gly-Tyr-Gly) located in the P-loop, which might inhibit its interaction with subunits of other K^+^ channels and K^+^ permeability [51,114,122,123,124,125]. *KCNV2*-related retinopathy had a similar genetic and phenotypic background across different ethnic group studies so far [126,127]. The genetic testing of an Arab cohort of 15 unrelated patients with a high prevalence of consanguineous marriage demonstrated the homozygous pathogenic variants in *KCNV2* in all the patients (Tyr53X in one patient, Glu143X in 13 patients, Cys177Ser in one patient) with delayed rod ERG b-wave response [128]. Likewise, biallelic loss-of-function compound heterozygous mutation (p.W67X and p.D174GfsX198) was observed as a cause of CDSRR in a Japanese patient with similar clinical traits whose unaffected parents harbored only one of the mutations [129]. A comprehensive molecular genetic analysis of a large Chinese cohort of 163 cone-rod dystrophy patients revealed that *KCNV2* mutations accounted for 0.6% of the cases [127]. In comparison, in the genotyping of 193 genes in a Japanese cohort of 43 patients, 2.3% of cases were *KCNV2* positive [126]. Recessive null mutations were reported ((Glu148Stop, His468fsX503, and Ala334fsX453) in the N-terminal of the protein and the P-loop leading to the production of a truncated protein with no difference in the CDSRR-phenotypic presentation [124]. The truncation or absence of K_v_8.2 protein leads to a positive shift in the photoreceptor membrane potential due to an increased intracellular K^+^ level and loss in K^+^ permeability. The K_v_2.1, without its assembly with K_v_8.2, does not generate a K^+^ outward current for regulating the photoreceptor response [130]. K_v_8.2-knockout mice have shown histological defects in ONL like reduced thickness and cell death, and functional anomalies were seen as severely reduced a-wave, delay in the b-wave generation, and absence of c-wave in response to light stimulation similar to CDSRR patients suggesting the channel functions are essential for vision [130,131].

### 2.6. BEST1 (Bestrophin-1; Ca^2+^ Dependent Cl^−^ Channel)

The *BEST1* gene containing 11 exons encode for the bestrophin-1 protein of 585 amino acids (NM_004183.4), expressed in the basolateral membrane of RPE [37,132]. *BEST1*- Promoter containing two E-boxes (−154 to −104 bps) is responsible for driving RPE specific expression by binding to *OTX2* and *MITF* transcription factors [133,134,135]. The BEST1 is a multifunctional protein with four transmembrane domains [136], which possibly oligomerizes to the dimer, tetramer, or pentamer [137,138,139]. The protein contains an intracellular N-terminal and a sizeable cytosolic domain with C-terminal. The channel’s Ca^2+^ hook (Ca^2+^ clasp), which binds to Ca^2+^ for its activation, consists of the helix-turn-helix motif in the first transmembrane domain (also known as EF1 region). An acidic cluster, one glutamic acid, and four aspartic acids are located adjacent to the last transmembrane domain [140]. Best1 functions as Cl^−^ channel gated by intracellular binding of Ca^2+^ [137,140,141,142,143,144,145,146,147,148,149], but it is also permeable to other monovalent ions like HCO_3_^−^, Br^−^, SCN^−^, I^−,^ and NO_3_^−^ [150,151]. Some of the studies showed volume regulated channel activity [139,152,153,154]. In the human retina, Ca^2+^ binding induces the conformational changes in the secondary structure of the Best1 channel, which is vital for its assembly and organization in RPE monolayer [155]. More than 250 *BEST1* gene mutations have been reported to date in different autosomal dominant (Best’s disease; BD, adult-onset vitelliform macular dystrophy; AVMD, Autosomal dominant vitreoretinochoroidopathy; ADVIRC, retinitis pigmentosa; *RP*), and an autosomal recessive (autosomal recessive bestrophinopathy; ARB) bestrophinopathies (HGMD database). The clinical presentation of autosomal dominant and recessive diseases with *BEST1* mutations has shown extensive variability in expression. Recessive mutations undergo a rapid ER-associated degradation while dominant mutation escapes this and suffer delayed endolysosomal lysis and, therefore, increased the stoichiometry of mutant versus normal subunits in BEST1 assembly [156]. Approximately 10% of the Best vitelliform macular dystrophy (BVMD) patients with *BEST1* mutations had normal vision and asymptomatic [157]. The rate of mutant protein degradation and subcellular quality control might be responsible for the distinct phenotype observed in dominant and recessive bestrophinopathies. Most of the observed mutations were missense and located at the N-terminal of protein. The N-terminus is evolutionary conserved and can severely affect the protein functions. A missense mutation, W93C located near to Ca^2+^ clasp region, abolished the channel function and Ca^2+^ homeostasis in RPE [158,159,160]. Some of the mutations located at N-terminus (T6P, V9M), a cytoplasmic loop between the second and third transmembrane domain (P101T, P152A, L174QfsX57, R200X) and C-terminal domain (V311G, D312N, V317M) reported in ARB, AVMD, and retinitis pigmentosa altered the channel trafficking leading to its accumulation in intracellular compartments rather than the basolateral membrane of RPE, as studied in Madin–Darby canine kidney (MDCK) heterologous system [161,162,163,164,165]. Apparently, few of the truncating mutations (L472PfsX10 and H490QfsX24) associated with ARB were properly localized in MDCK cells [163]. The channel also contains an adenosine triphosphate (ATP)-binding motif and BEST1 activity can be enhanced by ATP in a dose dependent manner. A missense mutation (p.I201T) located within this motif has shown diminished ATP-dependent activation of BEST1 in iPSC-RPE and corresponding mutants in *Best1* homologs showed defective ATP binding due to conformational changes in the motif [166]. ARB-associated mutations (R141H and I366fsX18) in iPSC-RPE reflecting the null phenotype exhibited reduced internalization of photoreceptor outer segment and impaired phagocytosis [167]. Mutations associated with bestrophinopathy have been studied in vitro and in vivo for their effect on the channel functions. The overexpression of W93C-BEST1 in human fetal RPE demonstrated the increase in tranepithelial electrical properties, the ablation of the Cl^−^ current, and altered Ca^2+^ signaling [159]. *Best1*-W93C homozygous and heterozygous mice, and *Best1*-knockout mice had normal Cl^−^ current but exhibited anomalous Ca^2+^ level in RPE altering the kinetics of voltage dependent Ca^2+^ channel which implies the significance of BEST1 as Ca2^+^ regulator [159,160,168]. 

### 2.7. KCNJ13 (Inwardly Rectifying K^+^ Channel)

The gene contains 3 exons that encode for the K_ir_7.1 protein of 360 amino acids (NM_002242.4). The K_ir_7.1 channel is a homo-tetrameric structure composed of an intracellular N- and C-terminal and two transmembrane domains linked by an extracellular pore-forming region [169,170]. In the eye, the channel is expressed in the apical membrane and microvilli of RPE and is crucial for regulating the subretinal space ionic homeostasis [171,172,173]. Kir7.1 is a weak inwardly rectifying potassium channel that controls the membrane potential in RPE. In light, the opening of the channel allows the efflux of K^+^ ion from RPE to the subretinal space to counterbalance the ionic concentration to its dark level. Studies have suggested the functional coupling of Kir7.1 with Na^+^/K^+^ ATPase in the apical membrane of RPE for the recycling of K^+^ ions [174]. Mutations in *KCNJ13* cause autosomal dominant SVD and autosomal recessive LCA16 [33,34,35,175,176]. SVD is characterized by fibrillary degeneration of vitreous humor, retinal detachment, and early-onset cataract, while LCA16 is characterized by nystagmus, photophobia, and complete vision loss [177,178,179,180]. A missense heterozygous mutation (c.484C>T, R162W) in *KCNJ13* in an SVD-patient showed a dominant-negative effect in *Chinese hamster ovary* (CHO-K1) cells. The molecular modeling of the mutant protein revealed the possible disruption of the channel structure. The mutant protein expression in CHOK1 produced a nonselective cation current leading to depolarization of the cells with increased fragility [34,176]. A homozygous nonsense mutation in *KCNJ13 (W53X)* was detected in an LCA16-patient of middle-eastern origin. The resultant truncated protein, when expressed in heterologous CHOK1 cells, did not traffic to the membrane, showed depolarization of membrane potential with an 85% reduction in the inward current. The siRNA mediated inhibition of Kir7.1 channel in mice showed reduced ERG waveforms, as observed in LCA16 patients [33]. Kir7.1 mutation has shown cellular defect in jaguar/Obelix zebrafish leading to change in the pigment patterns because melanosomes were not responsive to changes in the light [181]. Similar defects are observed in an LCA patient with compound heterozygous (c.314G>T; p.Ser105Ile and c.655C>T; p.Gln219X) mutation [182] and in a patient with vitreoretinal dystrophy with early-onset cataract which is homozygous for a missense mutation (c.359T>C; p.Ile120Thr) in *KCNJ13* [183]. These patients presented with clumpy pigment deposits in the macular area and notable fibrosis over the disc causing the dysfunctions and disorganization at the level of RPE. There are reports of G protein-coupled receptor (GPCRs) mediated glycosylation of Kir7.1 channel is required for its activity, and mutations in *KCNJ13* (L241P, Q117R) appeared to have complete loss or reduction in glycosylation, thereby altering the channel functions [184]. Various in vitro and in vivo models have shown that the Kir7.1 channel functions are required to maintain RPE and photoreceptor health. In vivo *KCNJ13* conditional or mosaic KO and knock-down mice exhibited a loss of photoreceptors and abnormal electroretinogram due to reduced or absent Kir7.1 [185,186,187]. A recent study in *KCNJ13*-KO hiPSC-RPE cells lacking Kir7.1 channel revealed the loss of phagocytic activity as well as the reduced expression of phagocytosis-related genes, which could be a possible cause of retinal degeneration as seen in LCA16 patients [188]. 

### 2.8. CNGA3 and CNGB3 (Cone Specific Cyclic Nucleotide-Gated Channel α and β Subunit)

The *CNGA3* gene contains 8 exons that encode for the α subunit of cone-specific cyclic nucleotide (CNG) gated cation channel comprised of 694 amino acids (NM_001298.3). The *CNGB3* gene with 18 exons encodes for the modulatory β subunit of CNG channel having 809 amino acids. CNG channel is heteromeric, comprised of three α and one β subunit [189]. The proper assembly (A3/B3; 3:1) of the CNG channel is essential for maintaining the protein integrity and trafficking to cone outer segments [190,191]. Unlike CNGB3, the homomeric CNGA3 assembly can also yield the functional channel, but its ligand selectivity and gating properties are different from that of the heteromeric A3/B3 channel [192]. Structurally, the α subunit is homologous to β, consist of six transmembrane domains, a pore-forming unit between the last two domains, and a C-linker, which connects the cyclic nucleotide-binding domain to the sixth transmembrane domain. The channel is a non-selective voltage-gated cation channel located in the cone outer segments, activated by binding to cyclic nucleotides (cGMP, cAMP) and plays a crucial role in phototansduction pathways [193,194]. Biallelic mutations in *CNGA3* and *CNGB3* genes are the most common cause of congenital autosomal recessive achromatopsia (ACHM) [195,196,197,198,199,200,201], which affects 1 in 30,000 individuals [202,203]. These patients manifest reduced visual acuity, photophobia, loss of cone functions, and reduced or complete loss of color discrimination, as shown by diminished or no ERG photopic (cone) response. The rod functions in these patients were preserved as reflected by healthy and stable scotopic (rods) ERG [195,204,205,206,207]. Extensive genotyping of *CNGA3/B3* genes documented more than 100 mutations in *CNGA3* and approximately 50 mutations in *CNGB3* with a significant fraction of missense mutations in *CNGA3* while nonsense mutations in *CNGB3* (HGMD database). Topological and structural modeling studies revealed that two missense mutations (c.1306C>T; p.R436W, c.1540G>A; p.D514N) in *CNGA3* reported in Pakistani patients with ACHM lead to the production of mutant proteins with abnormal structure affecting the C-linker and cyclic guanosine monophosphate-binding sites, respectively [201]. CNG channels are modulated by phosphoinositides (phosphatidylinositol 4, 5-bisphosphate (PIP_2_), and phosphatidylinositol 3,4,5-trisphosphate (PIP_3_) via C-terminal leucine zipper domain, which is vital for the channel assembly and intersubunit interactions. Mutations (e.g., *CNGA3*; L633P) in this region might impact the PIP sensitivity of the channels and contribute to disease phenotype [199]. A nonsense mutation in *CNGA3* (R23X) produced a shorter isoform, which cannot compensate for the functional loss of the long isoform hampering the assembly and transport of the CNG channel to the cone outer segment, subsequently leading to the loss of cone photoreceptor functions [208,209]. A canine model of *CNGA3*-R424W showed complete loss of cone function and channel activity by inhibiting the salt bridge formation while *CNGA3*-R424W exhibited the abnormal biogenesis of the trimeric channel [210]. Clinical heterogeneity of a *CNGB3*-missense mutation (c.1208G>A; p.Arg403Gln) was reported in patients with variable phenotypes like macular dystrophy/degeneration and cone dystrophy [196,211,212]. This mutation in the evolutionarily conserved pore helix of CNGB3 did not affect its heteromeric assembly with CNGA3 in Xenopus oocytes. Still, the channel showed increased ligand sensitivity and outward rectification [192,213]. Cone retinopathy-patients in Tubingen were found to harbor digenic-triallelic mutations in *CNGB3*/A3. Around 62.5% of the patients who had either homozygous (R403Q) or compound heterozygous mutations in *CNGB3* (R403Q+other mutation) also harbored an additional heterozygous mutation in *CNGA3* [214]. The *CNGA3* mutation was found to be pathogenic with a severe phenotype as compared to the patients who had only monogenic *CNGB3* mutations suggesting a hypomorphic effect of CNGB3-R403Q mutation. The animal studies further supported the role of digenic triallelic inheritance in cone retinopathies. The mouse model of digenic triallelic (*Cnga3^+/–^* Cngb3^R403Q/R403Q^) nature exhibited a severe disease phenotype as compared to *Cngb3^R403Q/R403Q^* mice as demonstrated by the loss of cone photoreceptor function and structural integrity [214].

### 2.9. CNGA1 and CNGB1 (Rod Specific Cyclic Nucleotide-Gated Channel α and β Subunit)

The *CNGA1* gene contains 11 exons that encode for the CNGA1 (α-subunit of CNG) protein of 690 amino acids (NM_001375386.1) While the *CNGB1* gene contains 33 exons which encode for the CNGB1 (β -subunit of CNG) protein of 1251 amino acids (NM_001297.5). CNGA1/B1 are rod specific cyclic nucleotide-gated (CNG) voltage-activated cation channels that exist as the hetero-tetrameric structure around an aqueous pore. Similar to the cone-specific CNG channel, rod-specific CNG channel is formed by three-A1 and one-B1 subunit. Each subunit is comprised of 6 transmembrane domains with a pore-forming loop (P-loop) between the fifth and sixth domain and a carboxy-terminal, which contains cyclic nucleotide-binding domain [5,193,194,215,216]. The channel opening is regulated by the binding of cyclic nucleotides and through a gating mechanism mediated via the fifth transmembrane domain of the channel [217]. The channel remains open in the dark after binding to a high concentration of cGMP and allows the influx of cations like Na^+^ and Ca^2+^ [218]. Homozygous mutations in *CNGA1* and *CNGB1* result in autosomal recessive retinitis pigmentosa (RP), a genetically heterogeneous group of retinal degenerative diseases with variable clinical phenotype, characterized by the degeneration of RPE and photoreceptors [42,219]. RP-patients present with typical symptoms like night blindness, visual field loss, optic disc atrophy, and diminished or no response in ERG. The screening of the *CNGB1* gene identified several deleterious missense, nonsense, frameshifts, and splice-site mutations (p.N986I, p.Q88X, p.Q222X, p.Q318X, p.R729X, p.A1048fsX13, p.L849AfsX3, c.761 + 2T>A) in RP-patients with RPE atrophy and intraretinal pigment migration [220]. A compound heterozygous mutation (G622A and D208N) was identified in *CNGA1* in a patient with autosomal recessive RP using a combined approach of NGS and sanger sequencing. In the human embryonic kidney 293T (HEK293T) heterologous expression system, these mutations resulted in the reduced expression of CNGA1 in the membrane, suggesting a possible mechanism of altered trafficking, leading to degenerated rod-phototransduction in the RP patient [221]. The screening for *CNGA1* in 173 unrelated patients with autosomal recessive RP reported one missense (Ser316Phe), one frameshift (Arg654fsX32), and two nonsense mutations (Glu76X and Lys139X) in 5 patients. Nonsense mutation encoded a protein lacking the essential channel domains. In contrast, missense and frameshift mutations produced a protein which predominantly retained in the cell instead of being trafficking to the membrane [42]. Another mutation (c.1537G>A; p.G513R) had a similar effect on the protein function suggesting the altered trafficking is deleterious to rod functions [222]. Downregulating the CNG channel using antisense RNA in mice showed deleterious effects on rods and degeneration of bipolar cells [223]. 

## 3. Disease Models Recapitulating Human Channelopathies

Inherited channelopathies caused by a spectrum of mutations are among the leading causes of blindness in the developed world. There is a need to characterize the gene mutations as they have their own role-play at the cellular and molecular level. Modeling the disease in an in vitro or in vivo system is the prerequisite to investigate the impact of mutations on molecular, biochemical, and biophysical properties of channel proteins. This will open the roads to elucidating the previously unknown disease mechanism and pathways underlying the specific channelopathies. Several studies have characterized the molecular abnormalities involved in the causation of channelopathies associated with different mutations. With the advent of technology, disease models are now available to test the safety and efficacy of gene and cell-based therapies safely and successfully translated into clinical use (Figure 6). These models have proven very efficient for the screening, identification, validation, and development of effective drugs to pioneer new therapeutic modalities. 

### 3.1. Engineered In Vitro Disease Models

Cell lines like HEK293, MDCK, and CHO-K1 have been extensively used to model the ocular channelopathies in vitro. These cell lines allow folding, trafficking, and required post-translational modification in proteins [224]. These are in vitro model to study the differences in heterologous expression (by immunocytochemistry, in-cell western analysis) and electrophysiological properties (by patch-clamp) of proteins between wild type (WT) and mutated protein. An LCA16-nonsense mutation in *KCNJ13* (W53X), when expressed in CHO-K1 cells, showed altered localization in the cytoplasm with no current as compared to a WT protein which is located in the cell membrane with a normal inwardly rectifying current. Also, the mutant protein did not alter the function of the WT channel suggesting the mutation does not cause any dominant-negative effect [33]. Similarly, an SVD-missense mutation (R162W) in CHO-K1 cells showed a 70% reduction in Kir7.1 protein expression as compared to the WT, which was further reduced to 50% when WT and R162W protein were co-expressed due to the oligomerization of WT with mutant protein, which might have affected the Kir7.1 assembly. R162W mutant protein also differed in its biophysical property. Cells with R162W mutant protein were depolarized with resting membrane potential nearly zero and had diminished amplitude of inward current with no preference for K^+^ or Rb^+^ [175]. A compound heterozygous *BEST1* mutation (R141H and I366fsX18) identified in autosomal recessive bestrophinopathy (ARB) patients was characterized in HEK293 cells. The co-expression of both the mutant protein resulted in rescued channel function with comparable Cl- current to WT, attributing ARB pathogenesis to the loss of the C-terminal domain in I366fsX18 mutant [152]. 

We rely on immortalized cell lines, cadaveric human eyes, and animal models to learn about the disease mechanism and explore the treatment modalities. Of late, iPSC modeling for human channelopathies has shown considerable advantage as it offers the potential to develop patient-specific cell lines ex vivo. These cells also circumvents the immunological problems associated with cell-replacement therapies. The possibility of iPSCs to differentiate into specific eye-specific lineages has provided an alternative tool to study disease pathophysiology, genotype–phenotype correlation, and drug testing. Patient-derived fibroblasts, reprogrammed to iPSC and further differentiated to RPE and photoreceptors, are morphologically, physiologically, and biochemically similar to the cells in vivo, provide a robust disease-in-a-dish model for various pathological conditions [225,226,227]. Best’s disease (BD)-specific iPSC carrying a missense mutation (c.888C>A, p.N296K) in the *BEST1* gene was established and characterized to be used as a disease model to investigate the pathogenesis [225]. There exist discrepancies between the heterologous system, like MDCK cells and iPSC-RPE. The mutant R141H-Best1 protein is mislocalized in MDCK cells, perhaps due to the overexpression of the mutant protein. In the patient iPSC-RPE, the R141H-mutant protein is trafficked to the membrane, suggesting a need to validate the native model [152,163]. Except for ARB, all other bestrophinopathies are the result of autosomal dominant mutations in *BEST1,* but available in vivo disease models have their limitations to characterize the dominant phenotype. A naturally occurring canine model has biallelic mutations leading to inherited retinal dystrophies and is used as a recessive disease model [228,229]. *Best1* heterozygous knockout (*VMD2*^+/−^) mice do not develop dominant mutation-associated phenotype for abnormalities in retina architecture and Cl^−^ current and cannot be used to characterize the dominant specific disease phenotype. [168]. These limitations of in vivo models were avoided by the development and characterization of iPSC-RPE with dominant genetic variation. An ADVIRC-specific iPSC-RPE harboring a missense mutation (p.V235A) showed mislocalization of the mutant protein to the apical and basolateral membrane. In contrast, WT is expressed primarily in the basolateral membrane of RPE [230]. Similarly, ARB-specific iPSC-RPE with a compound heterozygous mutation (R141H and I366fsX18) showed the mutant protein does not undergo NMD. These are expressed at the endogenous level with the impaired phagocytic ability [167]. 

### 3.2. Engineered In Vivo Disease Models

Animal studies have been incredibly useful for the development of effective and novel diagnostic and therapeutic interventions. An incomplete understanding of disease pathophysiology caused by different mutations limits our capability to find the treatment regimens. Disease models deciphering the exact phenotype and mechanism of pathophysiology is the first step for the design and testing of the therapeutic/pharmacological strategies. For example, *Trpm1*-null mice represented similar electrophysiological responses as cCSNB patients [60] and an excellent model to adopt. *Trpm1* long-form (Trpm1-L) in mouse is an orthologue of human *TRPM1 is* co-expressed with the glutamate receptor in the dendritic tips of the ON-bipolar cells [60]. *Trpm1^−/−^* mice exhibited a complete loss of ON bipolar cell response with reduced or no ERG b-wave amplitude in scotopic and photopic conditions, suggesting that the functional TRPM1 is required for signal transmission by the light-evoked response [60,231]. Likewise, *Kcnv2* knockout mice exhibited similar histological changes (reduced ONL, loss of rods, mild impairment of cones) and reduced ERG a-wave as noticed in patients with cone dystrophy with a supernormal rod response (CDSRR) having *KCNV2* mutations [130,131]. A knock-in mouse with homozygous W93C mutation showed the significant symptoms of BVMD like reduced electrooculogram light peak, lipofuscin accumulation in RPE, and subretinal hemorrhage [160,232]. The mouse model also confirmed the BVMD-disease etiology is due to Best1 dysfunction rather than its deficiency [160]. 

The prime requisite to study the effect of mutation is the homology of the gene sequence between humans and the disease model. The homolog of human CACNA1F in mouse (Cacna1f) has a 90% identity in amino acid sequence with near-perfect conservation between the functional domains. The functional characterization of the Ca_v_1.4α1 channel from mouse retina has shown that a slightly more negative voltage is required for human channel activation; otherwise, both the channels have consistency in their assembly with β subunit and DHP (dihydropyridine, the antagonist of Ca_v_1.4α1) sensitivity [233]. Therefore, mice lacking the *Cacna1f* gene could potentially be suitable in vivo model to study the disease mechanism of CSNB2. There are naturally occurring *Cacna1f* null mutant (*nob2*; no b-wave 2) mice with a defective synaptic transmission from photoreceptors to bipolar cells resulting in diminished scotopic and photopic ERG b-wave [234]. Further studies on these mice have shown that it is not a null mutant of Ca_v_1.4 channel. Despite the anatomical and functional defects in the retina, these mice had a normal vision based on the optokinetic response [235]. *Cacna1f^G305X^* is a complete knockout with degenerated rods and cones, absence of synaptic functions, and optokinetic response [236]. Although the loss-of-function mouse model of *Cacna1f* has established defective rod photoreceptor synapses, the reason behind cone degeneration in these mice remains unclear [91]. Zebrafish is another in vivo model more amenable for genetic manipulation to characterize the disease-causing mutations and develop personalized medicines timely. Zebrafish *wud* (wait until dark) mutants encode for a homolog of human Ca_v_1.4 channel (Ca_v_1.41a), 62.8% identical. The *wud* mutants are a unique tool to study the null effect of Ca_v_1.4 primarily in cones as they have cone-enriched retina, unlike the rod-dominated retina of mice with no fovea and, therefore, are candidate CSNB2 disease model. [237,238]. 

The remarkable genotypic, anatomical and physiological dissimilarities between humans and animals have prompted scientists to rely on more than one model for the complete understanding of disease etiology. To investigate the role of α2δ4, cone-dominant zebrafish knockout of two paralogs of *Cacna2d4* (*4a* and *4b*) proved to be an ideal model of pathophysiological mechanism due to *CACNA2D4* dysfunctions in human. This model not only determined its role in the formation of Ca_v_1.4-pore subunit and cone-mediated signal transmission but also confirmed its evolutionarily conserved function at the photoreceptor synaptic terminals [239]. Another reason to have multiple disease models is embryonic lethality caused by gene knockout, like in *Kcnj13* mice, which die at postnatal day 1 (P1) [240]. Due to these reasons, non-human primates like rhesus macaque have been of great importance. Although non-human primate (NHP) eye closely resembles humans, their use is limited due to limited availability and associated cost. A naturally occurring, non-human primate (NHP) model of achromatopsia with a homozygous mutation in the *PDE6C* gene had phenotypic similarities to that of the human achromatopsia patients and was used to understand the cone-mediated visual loss associated with CNGA3/B3 mutations. [241]. A canine model with two spontaneous mutations (R424W or V644del) mimicking the human *CNGA3*-associated achromatopsia has been shown a valuable system to study the cone-specific CNG channel function and for a proof-of-concept study of *CNGA3* gene therapy [210].

## 4. Potential Therapies for Reshaping the Non-Sensing Ocular Ion Channels to Sensing Ones

Genetic mutations contribute significantly to the stunning array of ocular channelopathies. The united efforts of clinicians and scientists enabled the discovery of the association of different ion channel genes to the disease phenotype. With the advances in technology, screening mutations across these genes and their effect on specific channel protein function have become possible. It is now known that these ion channels are multimeric in nature and function in-network with several other regulatory proteins. Research on the biophysical properties of ion channels has revealed the fundamental processes responsible for the selective nature of these proteins to understand the normal physiology. These details will help scientists provide a new promising paradigm for precise diagnosis of disease to facilitate the development of rational treatments. Several therapies (Figure 7) and clinical trials targeting the disease-specific genes (mutations), transcripts, or proteins in various ocular tissues are now in practice (Table 2). One advantage of testing the potential therapies in the eye is its easy accessibility for surgical intervention, non-invasive optical imaging, monitoring, and delivering different therapeutics. Also, the presence of the blood-ocular barrier makes the eye immune-privileged and limits the toxicity of drugs and therapies. Innovating the novel therapeutic approaches and their efficient delivery to the eye for various channelopathies is a viable treatment option. 

### 4.1. Therapeutic Approaches for the Correction of Genes

#### 4.1.1. AAV-Mediated Gene Therapy

Gene therapy validated for ocular ion channelopathies are recessive, monogenic, and caused by loss-of-function mutations [252,253]. It was also used to effectively knockdown a gene resulting in gain-of-function mutations [254]. Various adeno-associated virus (AAV) serotypes precisely deliver the therapeutic genes to retina layers, including photoreceptors, retinal pigmented epithelial cells, retinal ganglion cells, and bipolar cells either through subretinal or intravitreal injections. The non-viral mode of ocular delivery includes liposomes, nanoparticles, or direct DNA transfer. The therapy was tested for several ion channel genes in the eye like *TRPM1* [243], *BEST1* [244], *KCNJ13* [242], and *CNGs* [246,247,248,249,250,251] to provide a proof-of-concept for the treatment of diseases. The therapy aims to achieve the stable expression of the target gene without any conflict from the host immune system. An effective strategy to avoid the immune reaction is to use a specific gene promoter operating system to drive the tissue-specific expression of the target gene. The first gene therapy in the retina using AAV-based vector delivery of the *RPE65* gene was approved by the FDA to treat monogenic LCA2, which occurs due to recessive mutations [252,255,256]. AAV2 mediated subretinal *Best1* gene delivery (under the control of *VMD2* promoter) in a canine model (*Best1* biallelic mutations) of autosomal recessive bestrophinopathy (ARB) showed sustained and long term reversal of lesions and micro detachment in the retina, restoration of ONL thickness to normal values, and improved structure at RPE-photoreceptor interface without any inflammatory response in the eye [229]. The therapy has also been useful for the treatment of retinitis pigmentosa associated with *Cngb1* in mice. Subretinal delivery of normal gene using AAV2.1 (under the control of rhodopsin promoter) in *Cngb1*-null mice of autosomal recessive retinitis pigmentosa restored retinal morphology, CNGB1 channel expression, rod driven light responses in the retina, and delayed retinal degeneration [250,251]. The treatment also led to the restoration of the CNGA1 subunit, co-localized with CNGB1in rod of mice who performed better in vision-guided behavior tests than the untreated mice [250]. Gene therapy has shown success in the iPSC-RPE recessive disease model for the restoration of Kir7.1 and Best1 channel functions [242,244]. Mostly, recessive loss-of-function mutations are the prime target for gene therapy, but it is also capable of rescuing dominant loss-of-function mutations (*BEST1*; A10T, R218H, D302A, L234P, A243T, and Q293K) with similar efficacy in an iPSC-RPE model without the inactivation or disruption of the mutant allele [257]. The available data is very promising to launch the gene therapy for restoring the vision in patients with channelopathies. 

#### 4.1.2. CRISPR-Gene Editing

Over the past hundred years, genome editing breakthroughs have significantly revolutionized the disease modeling approaches. Among several programmable genome editing tools like meganucleases (MN), zinc finger nucleases (ZFN), and transcription activator-like effector nucleases (TALENs), the clustered regularly interspaced short palindromic repeat (CRISPR)-associated nuclease Cas9 (CRISPR-Cas9) is the most widely used editing tool because of its precise editing, efficiency, and ease of use in multiple modeling systems [258,259,260,261,262,263,264,265]. CRISPR element can be delivered to the eye as DNA (Cas9-sgRNA plasmid), mRNA (Cas9 mRNA, and sgRNA as two separate entities) or protein (Cas9 protein with sgRNA as ribonucleoprotein complex; RNP). Other than the viral-based in vivo delivery, nanoparticle-mediated delivery has been in use and considered to be safe [266,267,268,269]. The CRISPR can introduce as well as correct point mutations, insertions, and deletions in the cultured cells as well as in livings (mice, rats, monkeys, pigs, etc.) to reconstruct or correct the disease phenotype. The system edits the target gene by inducing targeted DNA double-strand breaks (DSBs) created by Cas9, which is directed to the target site by single-guide RNAs (sgRNAs) and protospacer-adjacent motifs (PAM, 5′-NGG-3′). DSBs activate the DNA-repair machinery of the cell for a defined genomic modification, which relies on exogenous single-stranded donor oligonucleotide (ssODN) template [270]. The gene-editing approach to repair DNA could benefit everyone with channel dysfunctions regardless of their mutation. The technique has the potential not only to correct missense mutations but also the nonsense and frameshift mutations. CRISPR-gene editing of autosomal dominant Best disease-associated missense mutations (*BEST1*; A146K, R218C and, N926H) in iPSC-RPE in vitro have shown highly efficient correction of the mutated allele with no alteration to WT allele. Single-cell patch-clamp for Ca^2+^-activated chloride current density showed the rescue of Best1 channel functions [244]. The approach is useful not only for gene corrections but also for creating models to understand the disease etiology. *Kcnj13* mosaic mice were generated by injecting CRISPR tools (guide RNA and Cas9) mimicking the LCA phenotype and helped dissect the gene function [186].

Similarly, to understand the *CNGA3*-associated achromatopsia, CRISPR elements were microinjected in zebrafish embryo to target *cnga3a* and *cnga3b* and disrupt the gene functions leading to impaired visual functions [271]. To generate *Cacna2df4*-KO mice, the α_2_δ-4 subunit function was disrupted by introducing a stop codon in the second exon using CRISPR-gene editing. The genetic silencing of α_2_δ-4 showed disrupted organization of rod synapses to a greater extent than cone synapses and abnormally extended horizontal and cone bipolar cell processes in the ONL correlating with a progressive loss of channel functions first in rods followed by cones. The study showed that the α_2_δ-4 channel maintains the integrity of photoreceptor synapses, the loss of which contributes to vision defects in patients with *CACNA2D4* mutations [100]. The CRISPR-Cas9 system is evolving continually for its better on-target efficiency and HDR in multiple cell types. One of the critical requirements for CRISPR-Cas9 system is PAM recognition, which can be overcome by engineering two different ScCas9 (Sc; *Streptococcus canis*) variants (Sc^++^ and HiFi-Sc^++^). These variants identify 5′-NNG-3′ PAM and therefore have broad genomic accessibility [272]. Similarly, two different SpCas9 (Sp; *streptococcus pyrogens*) variants, called SpG and SpRY, are developed, which eliminate the PAM dependency and can target most of the genomic loci with NGN- and NAN-PAMs. These variants are also compatible with the other PAMs (NCN or NTN) but with reduced efficiency [273]. Despite all these developments, several other concerns associated with the use of CRISPR remain unsolved: (1) Off-target effects associated with CRISPR due to non-homologous end joining (NHEJ) repair at DSBs, leading to lethal and undesirable permanent alterations in DNA; (2) the low efficiency of HDR in non-dividing cells like RPE, which are also deficient in active DNA repair machinery; and (3) monoallelic vs. biallelic correction.

#### 4.1.3. CRISPR-Base Editing

CRISPR-base editing, to a certain extent, addresses the concerns associated with gene editing. It uses a modified version of Cas9 (Cas9 nickase; nCas9) fused to either adenosine deaminase (A > G conversion) or cytosine deaminase (C > T conversion) along with sgRNA and a PAM sequence. These base editors are called adenosine base editors (ABEs) and cytosine base editors (CBEs), respectively. The system provides more specific editing with negligible unintended off-targets as nCas9 does not create a DSB and free of HDR/ active DNA repair system [274,275,276,277,278]. The approach is restricted to nonsense and missense mutations corrections as it mediates the change of a single “wrong “nucleotide to the “correct” one. Also, it enables only four nucleotide transitions (C > T, A > G, T > C, G > A) and does not apply to other transitions as well as frameshifts and indels. This makes it difficult to generalize and apply the technique to correct the vast majority of the mutated alleles in ion channel genes. However, is there a future for a universal and independent base editing, i.e., changing any base at the target? Emerging technological advancement has provided a more versatile and precise method “prime editing” to install the desired nucleotide change. It uses a nCas9 fused to reverse transcriptase domain and a specific guide RNA (Prime editing guide; pegRNA). The pegRNA is longer (>35bp) than a normal guide RNA(20bp) sequence as it contains the information to install it at a targeted location [279]. Prime editing complements the base editing by mediating the specific and efficient corrections of insertions, deletions, all possible twelve substitutions, and their combinations. Further research is mandatory to establish its potential in different cell types, and its in vitro and in vivo delivery. 

### 4.2. Therapeutic Approaches for the Correction of RNA (RNA Editing)

The use of CRISPR gene and base editing also depends on the presence of the PAM site at the target location, which can limit the correction of a few mutations. RNA editing or RNA editing for programmable A to I replacement (REPAIR) does not require a PAM site and, therefore, an alternative approach for the mutations which cannot be targeted by a PAM dependent Cas9. The A to I (adenosine to inosine) is the most usual RNA editing mediated by two adenosine deaminase acting on RNA (ADAR1 and ADAR2) [280,281,282,283,284]. Recently, the CRISPR-Cas13 system was used for editing RNA in a specific manner, which was later modified for its improved efficiency [285,286]. A modified form of Cas13 from *Prevotella* sp., which is catalytically inactive and cannot create DSBs fused to the ADAR2 deaminase domain, has shown higher specific RNA editing with undetectable off-targets [286]. As the RNA editing corrects the mutation in the transcript instead of the DNA of a defective gene, the changes made are temporary due to its short life span. The temporary changes made in RNA has its pros and cons. The advantage is, if there are any off-target edits, they will disappear over time as the RNA degrades eventually. However, at the same time, on-target edited RNA would also be unavailable to produce a corrected protein. This would lead to multiple-dose therapy for disease treatment. To date, RNA editing has not been applied for the treatment of any of the ocular diseases. Although this approach looks promising, it will take several years to be developed for clinical use in channelopathy patients.

### 4.3. Pharmacological/Small Molecules for the Correction of Proteins

#### 4.3.1. Readthrough Inducing Drugs

In-frame nonsense mutations are the most pathogenic mutations and, therefore, the potential target for the treatment of channelopathies. One promising and peculiar therapy for targeting the nonsense mutations is translational readthrough (RT) inducing drugs [287,288,289,290]. These are the small molecules like aminoglycoside antibiotics (AAGs), AAGs-derivatives (NB30, NB54, G418, NB74, NB84, NB122, NB124, etc.), Ataluren (PTC124), and its oxadiazole analogs which act as a nonsense suppressor [291,292,293,294,295,296,297]. These drugs are effective in in vitro and in vivo disease models of eye diseases like LCA16 and retinitis pigmentosa [242,298]. Nonsense mutations reflect the stop codon, UAG, UGA, and UAA in the mRNA transcript much earlier than the usual stop codon followed by PolyA. Once the ribosome reads these stop signals at its A-site (acceptor site), translation is terminated, leading to the production of a truncated protein with adverse or no functions. This results in the development of the disease phenotype. RT drugs can override these premature stop signals by tricking the ribosome to include a near-cognate amino acid resulting in a full-length protein, which can substantially circumvent the disease symptoms. RT drugs can interact with any of the factors (release factors (RFs), rRNA, mRNA, tRNA) participating in the translation of protein to hinder the interaction of RFs with stop codon or codon-anticodon recognition [299]. The treatment with RT drugs results in the insertion of near cognate amino acids at PTCs site with an ability to distinguish the standard stop signal to synthesize full length and functional protein. The UGA stop codon has shown the highest RT potential, followed by UAG based on the degree of codon termination fidelity [300]. The efficiency of RT drug depends on its biocompatibility with ocular structure. For example, NB30 and PTC124 work better than G418 and gentamicin in retinal cells and are less toxic [301,302]. The outcomes of read-through drugs also depend upon the availability of a stable transcript as mRNA carrying a PTC undergo NMD. Inhibiting NMD alongside RT drug treatment can restore the transcript abundance and, thus, the efficiency of readthrough. Several drugs like caffeine and SMG1 kinase inhibitors attenuate the NMD pathway [289,303,304,305]. To date, there is only one report available for RT suppression of nonsense mutation leading to channel dysfunction in which NB84 treatment of an LCA16 patient (*KCNJ13*; W53X) derived hiPSC-RPE rescued the Kir7.1 channel functions by incorporating a near-cognate amino acid at PTC site [242]. 

#### 4.3.2. Anticodon Engineered tRNA

RT drug has shown promising results in the correction of nonsense mutations in various ocular and non-ocular diseases. However, if any amino acid other than a WT amino acid is inserted at the PTC site, it might make a full length but non-functional protein due to misfolding, mislocalization, altered assembly, or result in a narrower channel that inhibits ionic flux across the membrane [306]. Recently, another approach to repair PTC has been developed, in which tRNA has been engineered via mutagenesis to identify and homologously replace the premature stop signals by incorporating a cognate amino acid. These tRNAs are engineered to be guided to disease PTC. Therefore, they have limited off-target effects. Anticodon edited tRNAs (ACE-tRNAs) is aminoacylated by endogenous aminoacyl-tRNA synthetase and delivered to ribosome by elongation factor 1-alpha for the readthrough of PTC [307]. The approach was used to rescue cystic fibrosis transmembrane conductance regulator (CFTR) protein-PTC (W1282X). This mutation cause cystic fibrosis (CF) and, when corrected using ACE-tRNATrp (engineered to insert W at PTC), results in the production of mature glycosylated CFTR protein (W1282W) localized to the plasma membrane [308]. The approach is productive in both in vitro and in vivo models and has broader applicability for the nonsense mutation associated channelopathies. 

#### 4.3.3. Proteostasis Regulators and Pharmacological Chaperons

Some of the missense mutations in ion channel genes result in the altered folding of channel proteins and affect its assembly to form the functional complex. Mutant proteins prone to misfolding can prompt the collapse of protein homeostasis (proteostasis), a pathway in which balance is maintained between protein biogenesis, folding, assembly, trafficking, aggregation, and degradation. Proteostasis regulators and pharmacological chaperones hold considerable promises to ease the effect of mutant proteins in channelopathies. Proteostasis regulators increase the capacity of proteostasis network components to achieve the optimum proteome in the cell for normal physiological functions. Pharmacological chaperones are small molecules that can penetrate the plasma membrane and bind to the target protein for its proper folding and trafficking to the membrane [309,310]. In ocular channelopathies, pharmacological chaperones were used in culture cells to restore the Best1 function. ARB-associated missense mutations (p.L41P, p.R141H, p.R202W, and p.M325T) observed in the *BEST1* gene were the suitable target for this approach as they do not alter the active site of protein but leads to the production of misfolded protein which undergoes a proteasome-mediated degradation (endoplasmic reticulum-associated degradation, ERAD). The proteasome inhibitor bortezomib (BTZ), along with a chaperone 4-phenylbutyrate (4PBA), could restore the expression of mutant protein, proper folding, partial localization to the cell surface, and rescue the Cl^−^ conductance similar to WT protein [245]. An in-depth understanding of the molecular mechanism leading to disease pathology is required to find the best possible target as the entry point for these drugs.

## 5. Concluding Remarks

Ocular ion channel proteins are essential for visual functions and processing. Although many important questions regarding stoichiometry and assembly of subunits along with biophysical and physiological functions of ion channel proteins have been answered, further studies are needed to address the impact of mutations in disease mechanism and pathophysiology to formulate the specific therapies. It is essential to ultimately translate all laboratory research-based therapies to clinics for the treatment of patients who have blindness caused by channelopathies. A combinational therapy or a cocktail of different genetic and/or pharmacological therapeutic agents at a nontoxic level could be a key for reversing the nonsense ion-channels to sense and restore vision.

## Figures and Tables

**Figure 1 ijms-21-06925-f001:**
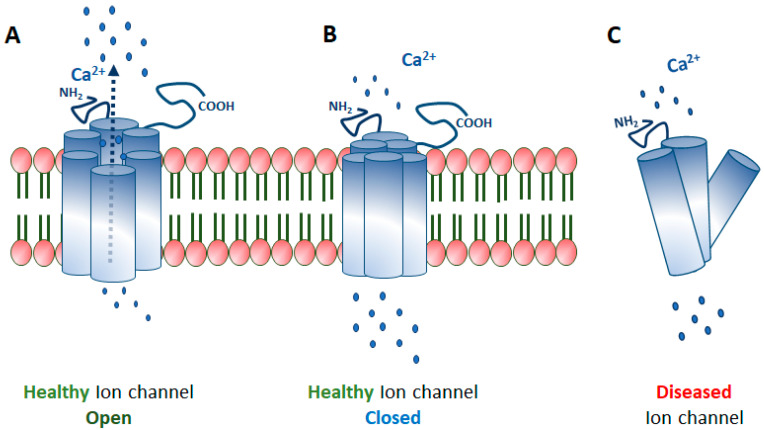
Ion channels; (**A**) An ion channel is open, allowing the flow of Ca^2+^ ions, (**B**) A closed ion channel limiting the flow of ions, (**C**) A truncated (diseased) ion channel not trafficked to the membrane due to dismantled subunits.

**Figure 2 ijms-21-06925-f002:**
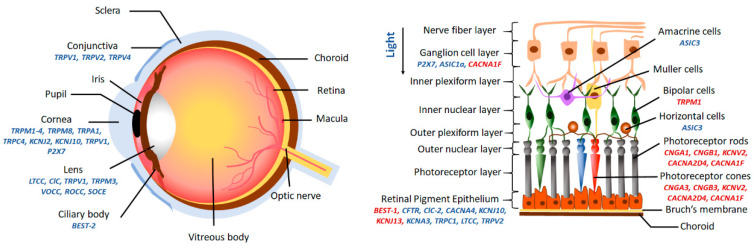
Anatomical location of ocular ion channels. Channels highlighted in the red cause associated blindness due to mutations.

**Figure 3 ijms-21-06925-f003:**
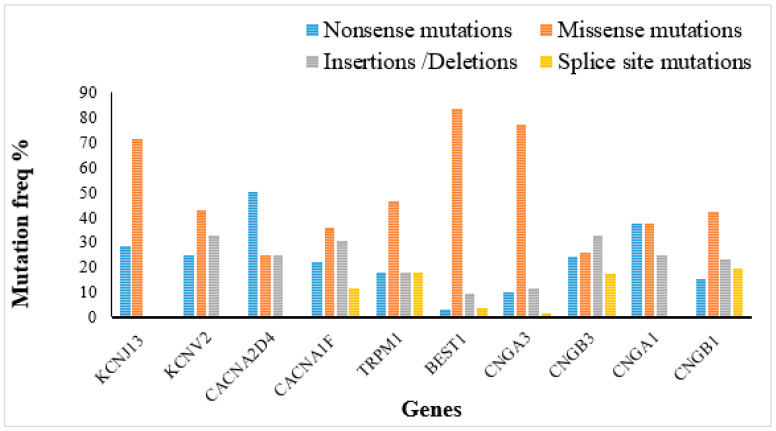
Distribution of disease mutations in ocular ion channel genes. (Based on HGMD and OMIM).

**Figure 4 ijms-21-06925-f004:**
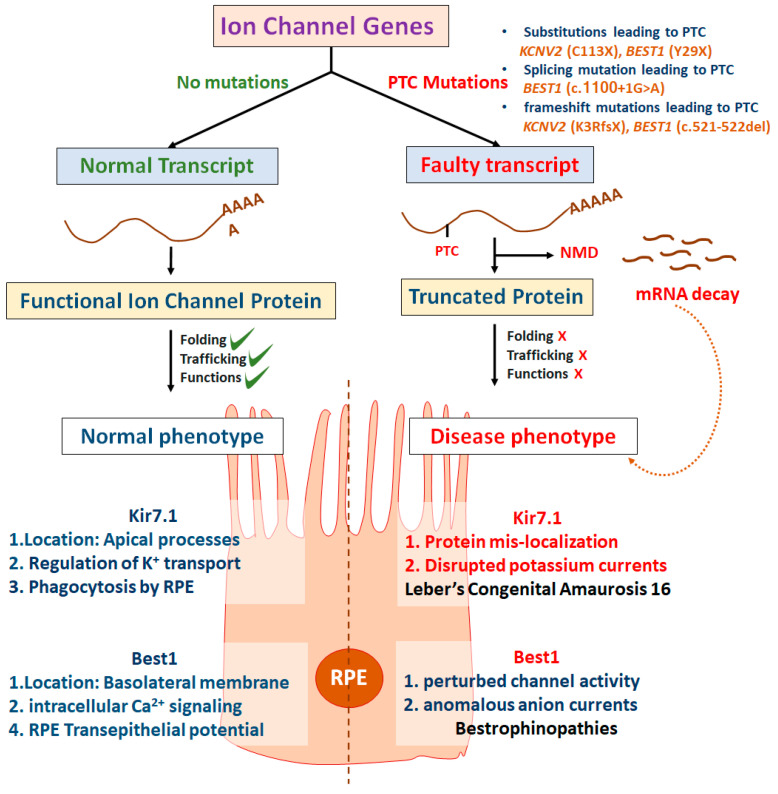
Leading nonsense mutation channel dysfunctions in the retinal pigmented epithelial (RPE) layer, causing genetic blindness.

**Figure 5 ijms-21-06925-f005:**
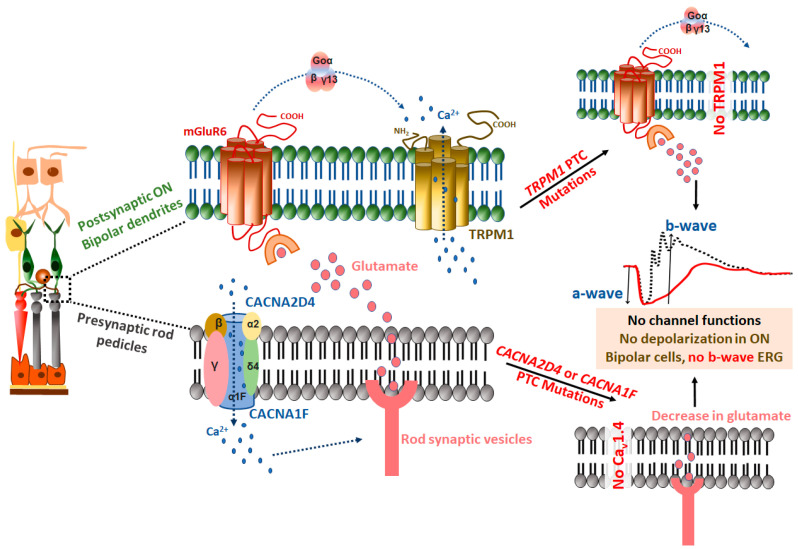
Nonsense mutations in ion channel genes underlying Congenital Stationary Night Blindness (CSNB). The electroretinogram (ERG) trace represents control (black trace) and no b-wave (red trace), a and b-wave are marked by downward and upward arrows, respectively.

**Figure 6 ijms-21-06925-f006:**
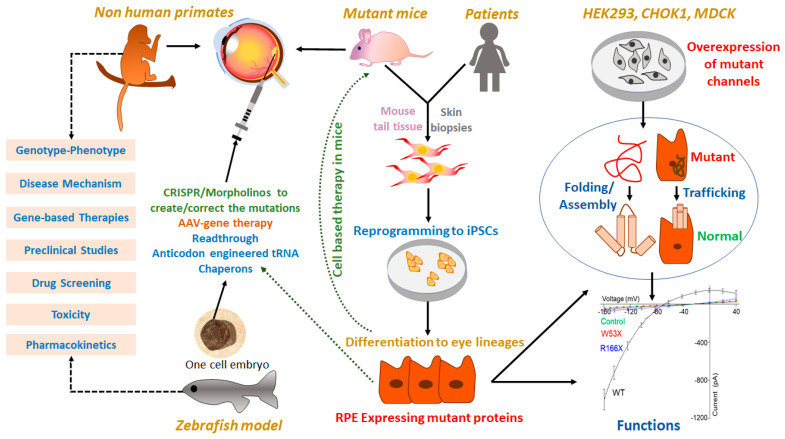
Disease models recapitulating human channelopathies and therapeutics.

**Figure 7 ijms-21-06925-f007:**
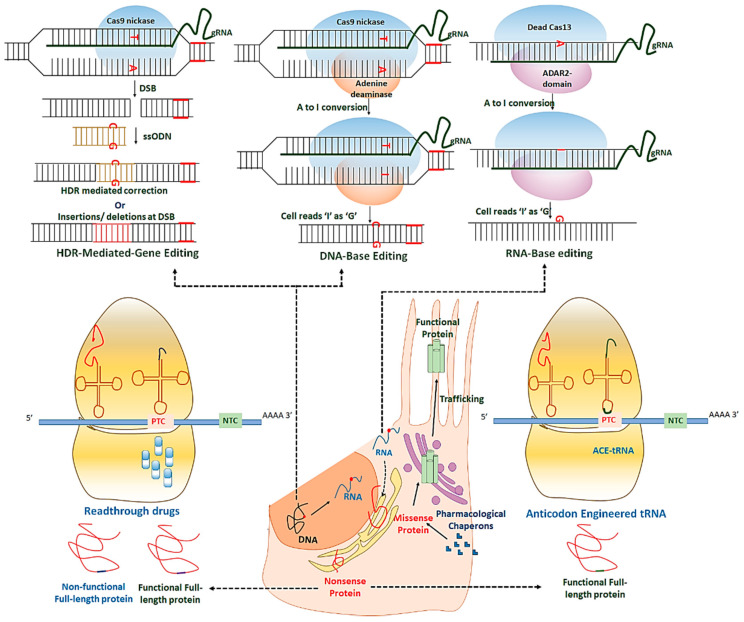
Potential therapies for ocular channelopathies caused by gene mutations.

**Table 1 ijms-21-06925-t001:** Nonsense mutations in the eye and resulting channelopathies.

Gene (#OMIM)	Location	Disease (OMIM)	Substitution Leading to PTC *
*KCNJ13* (#603208)	2q37.1	Leber congenital amaurosis-16 (#614186)	p.W53X, p.R166X
*KCNV2* (#607604)	9p24.2	Retinal cone dystrophy 3B (#610356)	p.K3X, p.W46X, p.Y53X, p.Y54X, p.E73X, p.Q76X, p.E80X, p.Q109X, p.C113X, p.E143X, p.Q145X, p.E148X, p.C177X, p.W188X, p.Q223X, p.K260X, p.Q287X, p.E306X, p.G461X
*CACNA2D4* (#608171)	12p13.33	Retinal cone dystrophy 4 (#610478)	p.R628X, p.Y802X
*CACNA1F* (#300110)	Xp11.23	Aland Island eye disease, Cone-rod dystrophy, X-linked 3 (#300600), Night blindness (#300476), congenital stationary (incomplete), 2A, X-linked (#300071)	p.R50X, p.R82X, p.E278X, p.Q325X, p.W349X, p.R379X, p.Q439X, p.R625X, p.R691X, p.R830X, p.R895X, p.R958X, p.R969X, p.R972X, p.R978X, p.S1114X, p.R1299X, p.Q1359X, p.W1451X, p.K1602X, p.R1827X
*TRPM1* (#603576)	15q13.3	Night blindness, congenital stationary (complete) 1C, (#613216)	p.Q11X, p.K294X, p.Y774X, p.W856X, p.R877X, p.S882X, p.E1032X, p.Y1035X
*BEST1* (#607854)	11q12.3	Bestrophinopathy (#611809), Macular dystrophy (#153700), vitelliform 2 (#193220), Microcornea, rod-cone dystrophy, cataract, and posterior staphyloma (#613194), Retinitis pigmentosa-50 (#613194), Retinitis pigmentosa concentric (#613194), Vitreoretinochoroidopathy (#193220)	p.Y5X, p.W24X, p.Y29X, p.K149X, p.R200X, p.W287X, p.R356X, p.S517X
*CNGA3* (#600053)	2q11.2	Achromatopsia 2 (#216900)	p.S21X, p.R23X, p.W171X, p.Q196X, p.R221X, p.W316X, p.E344X, p.W358X, p.W440X, p.R499X, p.Q537X, p.Q655X, p.K659X
*CNGB3* (#605080)	8q21.3	Achromatopsia 3 (#262300), Macular degeneration (#248200)	p.Q38X, p.Q131X, p.R203X, p.R216X, p.W234X, p.E336X, p.R355X, p.W373X, p.Y398X, p.E419X, p.R478X, p.W487X, p.Y545X, p.Q556X
*CNGA1* (#123825)	4p12	Retinitis pigmentosa 49 (#613756)	p.R32X, p.C39X, p.E80X, p.K143X, p.L174X, p.R424X, p.R514X, p.R560X, p.R629X
*CNGB1* (#600724)	16q21	Retinitis pigmentosa 45 (#613767)	p.C632X, p.Y787X, p.Y836X, p.W920X

* Mutation data is adapted from HGMD (http://www.hgmd.cf.ac.uk/ac/index.php) and OMIM (https://www.ncbi.nlm.nih.gov/omim).

**Table 2 ijms-21-06925-t002:** Therapies for ion channelopathies in practice and clinical trials.

Gene	Therapy In Vitro/In Vivo	References
*KCNJ13*	AAV gene therapy in iPSC-RPE in vitro	[242]
	Read through in iPSC-RPE in vitro	[242]
*TRPM1*	AAV gene therapy in mice	[243]
*BEST1*	AAV2-gene therapy in dogs	[229]
	Gene augmentation and CRISPR-gene editing in iPSC-RPE	[244]
	Pharmacological chaperons in culture cells (MDCK/HEK293)	[245]
*CNGA3*	AAV5-mediated gene therapy in mice AAV5-mediated gene augmentation in sheep	[246,247]
*NGB3*	AAV-gene therapy in mice	[248,249]
*CNGB1*	AAV-gene therapy in mice	[250,251]

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
