# Peer review of "Sensing through Non-Sensing Ocular Ion Channels"

_ijms, 2020, doi:10.3390/ijms21186925_

Round 1
Reviewer 1 Report
This is a thorough review of ocular channelopathies by an emerging leader in the field. The manuscript is focused on channelopathies specifically associated with diseases of photoreceptors. There are a few grammar/usage errors peppered within the paper as to be expected with such an effort. Manuscript is very thorough to the point of being unduly dense. The paper could use careful editing to be more succinct and concise.
Minor usage and grammar notes:
line 14, overserved
line 41, identify fig number
line 45, epic? usage?
line 63, be consistent with abbreviations ie. Ca2+
line 66, fig number
line 98, extra /
line 149, "get" depolarized instead say are PR cells depolarize....
line 151, "on the expense..." change to "at the expense..."
line 183 extra space
line 281, replace till with to
line 327-330, remove italics
Author Response
We thank the reviewers for their time in diligently reviewing this manuscript and providing positive feed-back. We have made all changes as suggested and the manuscript reads much better now. Please find all suggested changes through our track change and provided table.
Previous line No |
Suggested changes |
New line No |
line 14 |
overserved |
line 14 |
line 41 |
identify fig number |
line 42 |
line 45 |
epic? usage? |
line 46 |
line 63 |
be consistent with abbreviations ie. Ca2+ |
line 65 |
line 66 |
fig number |
line 70 |
line 98 |
extra / |
line 102 |
line 149 |
"get" depolarized instead say are PR cells depolarize.... |
line 153 |
line 151 |
"on the expense..." change to "at the expense..." |
line 155 |
line 183 |
extra space |
line 192 |
line 281 |
replace till with to |
line 290 |
line 327-330 |
remove italics |
line 337-340 |
Reviewer 2 Report
Kabra and Pattnaik have done a great job by nicely summarizing the ocular channelopathies associated with mutations in several different ion channels and also discussed potential therapies.
Please consider the below minor comments:
- The title should be adapted to include "ocular ion channels".
- An introduction to electroretinogram a, b and c waves with a figure will prove helpful to a broader audience, given that this is discussed at multiple places in the review article.
- Line 51 - Reference #11 seems inappropriate and rather a reference should be cited which describes the relevance of the mentioned channels in maintaining Ca2+ homeostasis in the lens.
- Line 637 - point no. (3) Recently some flexibility in PAM has been reported (see Walton et al., Science 2020 and Chatterjee et al., Nat. Biotechnol. 2020) which might be worth mentioning briefly.
Please also fix the following:
- Line 14 - "overserved" should be changed to "observed".
- The number in the legend of Figure no. 1 is missing.
- Line 52 - RPE should be expanded when using for the first time.
- Line 98 - "assembly/ /trafficking/function" should be changed to "assembly/trafficking/function".
- The number in the legend of Figure no. 2 is missing.
- Line 159 - ERG should be expanded when used for the first time.
- Line 227 - OPL and ONL should be expanded when using for the first time. Also please consider to show the OPL and ONL in figure 2.
- Line 245 - "K+" should be changed to "K+".
- Line 249 - "in once patient each" should be rephrased.
- BVMD which is introduced in the text in line 507 should already be introduced in line 288.
- Line 312 - Please change "Ca2+" to "Ca2+"
- Lines 327-330: Why Italicized?
- Line 345 - "model has shown" to be changed to "models have shown".
- Line 396 - "exhibited the exacerbated the" needs to be rephrased.
- Lines 479-487: Why highlighted?
- Line 538 - "P1" should be expanded to "postnatal day 1 (P1)".
- Line 585 - "RPE-PR interface" should be expanded to "RPE-photoreceptor interface".
- Line 728 - "Protein" should be changed to "protein".
- Lines 730 and 735: "chaperons" should be changed to "chaperones"
Author Response
We thank the reviewer for the detailed comment which we have now addressed. The manuscript now is edited for language and all the recommended changes. We feel the manuscript reads much better now. We have included a table to direct to changes made and new line numbers.
- The title should be adapted to include "ocular ion channels".
Response: The title has been changed to “Sensing through Non-Sensing Ocular Ion Channels”
- An introduction to electroretinogram a, b and c waves with a figure will prove helpful to a broader audience, given that this is discussed at multiple places in the review article.
Response: Along with the insertion of text details of a- and b-wave of ERG in section 2.2, an ERG figure showing the normal ERG and no b-wave is now included in figure 5.
- Line 51 - Reference #11 seems inappropriate and rather a reference should be cited which describes the relevance of the mentioned channels in maintaining Ca2+ homeostasis in the lens.
Response: The reference has been edited to the relevant one.
- Line 637 - point no. (3) Recently some flexibility in PAM has been reported (see Walton et al., Science 2020 and Chatterjee et al., Nat. Biotechnol. 2020) which might be worth mentioning briefly.
Response: The suggested studies have been included in section 4.1.2 (647-655 lines, references 276 and 277).
Previous line No |
Suggested changes |
New line No |
Line 14 |
"overserved" should be changed to "observed". |
Line 14 |
Line 41 |
The number in the legend of Figure no. 1 is missing. |
Line 42 |
Line 52 |
RPE should be expanded when using for the first time. |
Line 54 |
Line 98 |
"assembly/ /trafficking/function" should be changed to "assembly/trafficking/function". |
Line 102 |
Line 68 |
The number in the legend of Figure no. 2 is missing. |
Line 70 |
Line 159 |
ERG should be expanded when used for the first time. |
Line 163 |
Line 227 |
OPL and ONL should be expanded when using for the first time. Also please consider to show the OPL and ONL in figure 2. |
Line 236-237 |
Line 245 |
"K+" should be changed to "K+". |
Line 254 |
Line 249 |
"in once patient each" should be rephrased. |
Line 258-259 |
Line 288, 507 |
BVMD which is introduced in the text in line 507 should already be introduced in line 288. |
Line 297, 518 |
Line 312 |
Please change "Ca2+" to "Ca2+" |
Line 322 |
Lines 327-330 |
Why Italicized? |
Line 337-340 |
Line 345 |
"model has shown" to be changed to "models have shown". |
Line 355 |
Line 396 |
"exhibited the exacerbated the" needs to be rephrased. |
Line 406-407 |
Lines 479-487 |
Why highlighted? |
Line 490-498 |
Line 538 |
"P1" should be expanded to "postnatal day 1 (P1)". |
Line 551 |
Line 585 |
"RPE-PR interface" should be expanded to "RPE-photoreceptor interface". |
Line 598 |
Line 728 |
"Protein" should be changed to "protein". |
Line 750 |
Lines 730 and 735 |
"chaperons" should be changed to "chaperones" |
Line 752, 757 |